# Understanding depression treatment and perinatal service preferences of Kenyan pregnant adolescents: A discrete choice experiment

**Manasi Kumar**[1,2]*, **Albert Tele**[3,4], **Joseph Kathono**[1,5], **Vincent Nyongesa**[1], **Obadia Yator**[1], **Shillah Mwaniga**[3,5], **Keng Yen Huang**[6], **Mary McKay**[7], **Joanna Lai**[8], **Marcy Levy**[8], **Pim Cuijpers**[3], **Matthew Quaife**[9], **Jurgen Unutzer**[10]

1 Department of Psychiatry, University of Nairobi, Nairobi, Kenya, 2 Brain and Mind Institute, Aga Khan University, Nairobi, Kenya, 3 Vrije University, Amsterdam, Netherlands, 4 Ikuze Africa, Nairobi, Kenya, 5 Nairobi Metropolitan Services, Nairobi, Kenya, 6 New York University Medical School, New York, New York, United States of America, 7 Washington University in St. Louis, St. Louis, Missouri, United States of America, 8 UNICEF Headquarters, New York, New York, United States of America, 9 London School of Tropical Medicine and Hygiene, Bloomsbury, United Kingdom, 10 University of Washington, Seattle, Washington, United States of America

* manasi.kumar@aku.edu

**Data Availability Statement:** The datasets generated and/or analyzed during the current study

## Abstract

### Background

Understanding mental health treatment preferences of adolescents and youth is particularly important for interventions to be acceptable and successful. Person-centered care mandates empowering individuals to take charge of their own health rather than being passive recipients of services.

### Methods

We conducted a discrete choice experiment to quantitatively measure adolescent treatment preferences for different care characteristics and explore tradeoffs between these. A total of 153 pregnant adolescents were recruited from two primary healthcare facilities in the informal urban settlement of Nairobi. We selected eight attributes of depression treatment option models drawn from literature review and previous qualitative work. Bayesian d-efficient design was used to identify main effects. A total of ten choice tasks were solicited per respondent. We evaluated mean preferences using mixed logit models to adjust for within subject correlation and account for unobserved heterogeneity.

### Results

Respondents showed a positive preference that caregivers be provided with information sheets, as opposed to co-participation with caregivers. With regards to treatment options, the respondents showed a positive preference for 8 sessions as compared to 4 sessions. With regards to intervention delivery agents, the respondents had a positive preference for

are anonymized and shared via deidentified excel sheet as a Supporting information file.

**Funding:** Research reported in this publication was supported by the Fogarty International Center of the National Institutes of Health under Award Number K43TW010716-05. The content is solely the authors' responsibility and does not necessarily represent the National Institutes of Health's official views. The first author was funded by the Fogarty Foundation K43 grant (2018-2023), and the co-authors are her mentors and collaborators in this study. The funders had no role in study design, data collection and analysis, decision to publish, or preparation of the manuscript.

**Competing interests:** The authors have declared that no competing interests exist.

facility nurses as compared to community health volunteers. In terms of support, the respondents showed positive preference for parenting skills as compared to peer support. Our respondents expressed negative preferences of ANC service combined with older mothers as compared to adolescent friendly services and of being offered refreshments alone. A positive preference was revealed for combined refreshments and travel allowance over travel allowance or refreshments alone. A number of these suggestions were about enhancing their experience of maternity clinical care experience.

## Conclusion

This study highlights unique needs of this population. Pregnant adolescents' value responsive maternity and depression care services offered by nurses. Participants shared preference for longer psychotherapy sessions and their preference was to have adolescent centered maternal mental health and child health services within primary care.

## Introduction

The prevalence of depression is high among pregnant women, with worldwide estimates of 11–18% [1, 2] and between 15–28% in Lower-and-Middle-Income Countries (LMICs) [3, 4]. Adolescent mothers usually experience higher rates of prenatal depression as compared adult mothers [5]. Maternal depression negatively impacts the maternal and child health [6–8]. In Kenya, pregnant adolescents report mental health problems, difficulty in accessing financial, moral and material support from parents or partners, and stigmatization by health workers when seeking health care [9]. Discrete choice experiments (DCE) enable us to estimate relative preference weights and their corresponding trade-offs to measure what is important to people when choosing to engage in care [10], and though this approach has been tested within mental health field [11], it can more actively be used in low resource contexts to prioritize patient centered care [12]. We have done several studies that map epidemiology of mental disorders and risk factors experienced by pregnant and parenting adolescents along with rich qualitative work. The current DCE study is a step forward to elicit pregnant girls' treatment and care preferences so that improved interventional designs can be developed [13–16].

DCEs offer rigorous and systematic approaches for eliciting preferences for service or product attributes from customers and stakeholder [17]. DCEs allow for estimation of the relative importance of aspects of the service by analyzing trade-offs between attributes made by stakeholders. This method is increasingly applied to healthcare settings to enable patient input for patient-centered care [18] and has been successfully applied for patient preference elicitation in multiple areas of healthcare, including provider-interactions, health service delivery content and format, and treatment options [13, 18–21].

Developing innovative and scalable interventions to reduce the adverse consequences of depression in pregnant adolescents most of which emanate from social and interpersonal ostracization fundamentally require bolstering their engagement through offering adolescent friendly choices and services in the community and health care system. Pregnant adolescents are more likely to report symptoms of depression, stress, stigma, discrimination, isolation, marginalization, and suicidal ideation and this is further aggravated when there are intersections of Human Immunodeficiency Virus (HIV), tuberculosis (TB), disability or any other kind of severe adversities. Patients' preferences are particularly salient in depression treatment,

because multiple efficacious treatments (for example, combination of antidepressants and psychotherapies) and modalities (for example, group and individual) as well as different types of psychotherapies (cognitive-behavioral or relational like interpersonal psychotherapy (IPT)) exist. Incorporating individual patients' preferences into treatment decisions could lead to improved adherence to treatments for depressive disorders in this highly vulnerable group. We have tested IPT specifically and are developing adaptations of IPT [14, 22–24].

The objective of conducting this DCE was to quantitatively measure adolescent depression treatment preferences for different care characteristics and explore tradeoffs between these.

## Methods

### Design

A DCE is a survey design that allow participants to be estimated from their choices [25]. The method is based on random utility theory [26] and Lancaster's economic theory of value [27]. It is built on the assumptions that health care interventions, services, or policies can be described by their characteristics (called attributes), and that a person's valuation depends on the levels of these characteristics [28, 29].

DCEs ask people to complete a series of hypothetical choice activities to extract this information. Individuals are asked to choose their most preferred service among two or more alternatives (e.g., psychological therapies) with varying intervention characteristics in each choice task. Patient preferences can thus be measured as the extent to which each intervention feature influences an individual's intervention choice.

This survey was created in accordance with the International Society for Pharmacoeconomics recommendations and the Outcomes Research (ISPOR) Conjoint Analysis Task Force checklist for appropriate research techniques for stated-preference studies [30]. The task force's experimental design guidelines [31] were used in this investigation (see S1 File).

To investigate depression treatment preferences among pregnant teenagers, we employed a DCE, which consists of four stages: identifying and defining attributes and levels, generating choice sets and constructing questionnaires, collecting survey data, and analyzing and explaining the results [27, 32]. Fig 1 depicts the DCE's development process.

### Attributes and levels

A preliminary list of attributes was made by extracting all relevant attributes and levels from health related DCEs [33, 34] and through comprehensive literature review in consultation of with psychologists with experience in adolescent mental health. We supplemented this by conducting semi-structured qualitative interviews with experts in the field of adolescent mental health and health economics (primary health clinicians, nurses, and mental health practitioners (n = 36), researchers in the field of mental health (n = 10), and a health economist) [35, 36]. Subsequently, we conducted qualitative semi-structured interviews with 10 purposefully sampled respondents with a history of depression diagnosis. Following a grounded theory approach in the phases of both data collection and analysis [37], we derived lists of actors and factors that may play a part in the search for and selection of depression treatment. The authors reduced the set of potential candidate attributes to a more manageable set of attributes by filtering out double or overlapping attributes. The final list included eight attributes that consisted of; (i) Information delivery, (ii) Participants, (iii) Treatment option, (iv) Intervention delivery, (v) Training, (vi) Support, (vii) Services and (viii) Incentives.

**Attributes and levels description.** Eight attributes that consisted of treatment benefits were selected. The first one was on *Information delivery (two levels- Health facility, individualized)* where we asked participants if they would like information on how to use services in the

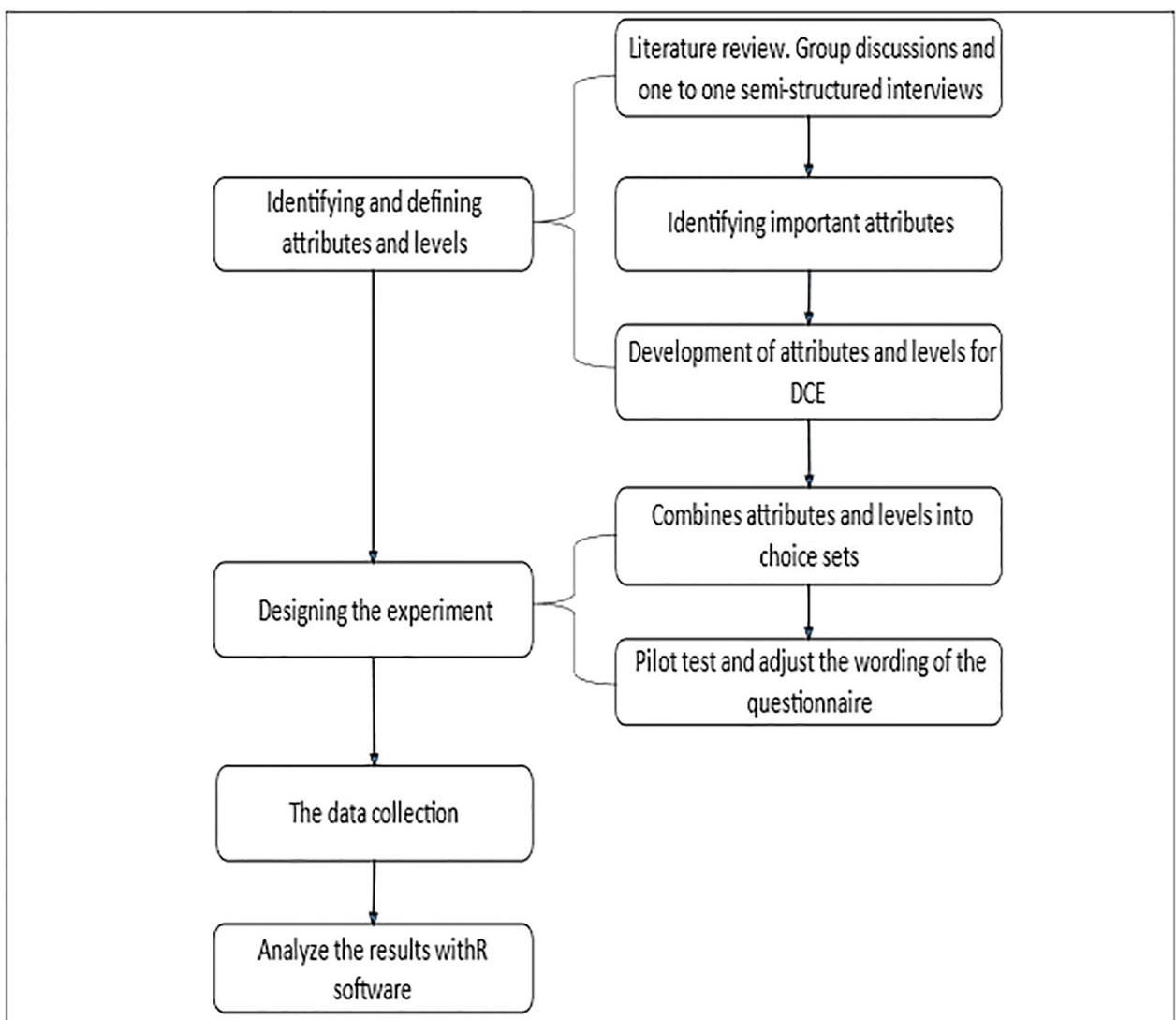

**Fig 1. The development process of DCE.**

facility around SRHR and MCH that could assist with depression management and the second level focused on provision of individualized information sheets on general care including nutritional care for mother and baby. The second option focused on *Additional Participants (Co-participate with Caregivers, provide information sheets for care-givers)* included asking if adolescents would like group sessions with their caregivers including partners vs information sheet on depression care for their partners. The third choice was around *Treatment duration option (4 sessions for 1.5 hours, 8 sessions for 1.5 hours)* for the entire depression treatment. The fifth choice was around *Intervention delivery agents (whether CHV, or Facility nurses)* probing if the therapy for depression should be delivered by lay workers or facility -based nurses. We also included an attribute around additional training support (exploring links to vocational training vs more formalized learning needs (i.e. back to school) were tested. We also had *a choice task around further support needs so we asked whether greater peer support or parenting skills support were preferred)*. The seventh choice in terms of *rethinking the health services*

**Table 1. Attributes and levels.**

| Attributes | Level |
|---|---|
| **1. Information delivery** | Health Facility (Ref.) |
| | Individualized |
| **2. Participants** | Co-participate with Caregivers (Ref.) |
| | Provide information sheets for care-givers |
| **3. Treatment option** | 4 sessions for 1.5 hours (Ref.) |
| | 8 sessions for 1.5 hours |
| **4. Intervention delivery** | CHV (Ref.) |
| | Facility Nurses |
| **5. Training** | Vocational (Ref.) |
| | Formal (Back to school) |
| **6. Support** | Peer support (Ref.) |
| | Parenting skills |
| **7. Services** | Adolescent friendly services (Ref.) |
| | Combined with older mothers |
| **8. Incentives** | Transport KSh. 500 |
| | Food |
| | Both (Ref.) |

*(Adolescent friendly services, combined with older mothers)* the choice given was between using and strengthening adolescent friendly services or using the routine MCH services and be serviced along with older adult mothers and finally in order to address the challenges that *Incentives* for improving health care uptake choices were offered *between (transport allowance, refreshment provision, or a choice for provision of both).*

For this analysis we applied dummy coding where the baseline attribute category for each attribute is omitted from estimations, and used as a reference category. The reference categories applied in this study are indicated in Table 1 as reference.

The design was pilot-tested with a selection of the pregnant adolescents who had been participated in the qualitative interviews to refine the survey and to assess the salience of the attributes to the treatment decision. Participants completed DCE questionnaires and participated in a personal cognitive interview as part of the pilot testing. To determine the burden on participants, the number of completed items and the time it took to complete them were recorded. Personal cognitive interviews were utilized to assess participants' knowledge of the questionnaire's levels and face validity. The final set of attributes and levels are presented in Table 1.

We tested multiple-choice elicitation formats and chose to use full-profile tasks between two treatment profiles in which participants indicated which treatment they would prefer to take. This setup allowed for the elicitation of acceptable tradeoffs people were willing to make between different treatment attributes. If the number of attributes is low enough that participants can reasonably complete a full-profile task, this maximizes information about trade-offs [38]. We allowed the participants to select an opt-out option. An example choice task with decision scenario is shown in Fig 2.

## Experimental design

We piloted using a fractional factorial design, then analyzed data in a multinomial logistic regression model to generate a Bayesian D-optimal design [39]. D-optimal designs maximize the precision of the estimated parameters given a set number of choice tasks and information

| Choice card 2 | | | |
|---|---|---|---|
| Attributes | Option A | Option B | Opt out |
| 1) Information delivery | Individualized dietary information mother and baby | Sexual reproductive health knowledge | None |
| 2) Caregiver & male partner participation | In-person participation of caregivers and male partners in session | Provide written reading materials on mental health to caregivers & male partners | None |
| 3) Treatment option | 8 sessions for 1 hour, 30 minutes | 4 sessions for 1 hour, 30 minutes | None |
| 4) Intervention delivery | Facility nurses | CHV (Community health volunteers) | None |
| 5) Support | Peer support | Mentorship/ support from older people | None |
| 6) Education & training | Return back to school (Formal training & certification) | Vocational training (Practice life skills for income generating schools) | None |
| 7) Services | Adolescent friendly services | Combined with older mothers | None |
| 8) Incentives | Transport | Food | None |
| Your choice? (Please tick one box) | ☐ | ☐ | ☐ |

**Fig 2. Sample choice card using orthogonal design.**

on expectations of respondent preferences [40]. We designed ten choice tasks, attribute levels were assumed to be categorical and uniformly distributed. A repeat task was added to test choice consistency. This design was sufficient to estimate main preference effects without interactions between attributes and was sufficient to answer our research question. In addition to discrete choice tasks, participants were asked to directly rank the attributes in order of importance from 1–8.

### Recruitment and data collection

Purposive sampling was used to recruit pregnant adolescents aged 14–18 years who were seeking antenatal services at two primary health care centers located in two informal settlements with Nairobi County were recruited from March to June 2022. In addition, respondents who

had participated in the preliminary study were approached (qualitative interviews and pilot study). The recruitment was carried out by two research assistants and seven CHVs. In Addition, 11 choice scenarios, we administered PHQ-9 to measure self-reported depressive symptoms and also collected information on their socio-demographic profiles. Confirmed pregnancy status, adolescent age 14–18 years, willingness to share their feedback on the DCE, familiarity with Kiswahili and English languages, stable mental health, being in the neighborhood for last one year, willingness to give consent were the exact inclusion criteria.

The sample size estimate in our study is based on Johnson and Orme's rule of thumb [41]. The calculation formula for the minimal sample size N, according to Johnson and Orme, is provided in the following equation: $N \geq (500 \times c)/(a \times t)$—where N is the number of participants, t is the number of choice tasks (questions), a is the number of alternative scenarios and c is the largest number of attribute levels for any one attribute, and when considering two-way interactions, 'c' is equal to the largest product of levels for any two attributes—(500 x 6/ 3 x 8). To account for 10% non-response at least 139 participants is recommended. A tablet-based DCE interview using Dooblo software program [42] was used to capture participant responses.

**Ethics approval and consent to participate.**   The study was approved by the Kenyatta National Hospital/University of Nairobi ethical review committee (approval no. P694/09/ 2018). The study received approval from the Nairobi County health directorate (approval no. CMO/NRB/OPR/VOL1/2019/04) and approval from National Commission for Science, Technology, and Innovation (NACOSTI/P/19/77705/28063). All study participants' informed consent to participate would be sought, including stakeholders and advisory committee members from whom data would be collected. The research was carried out per the KNH/UoN ethical review committee guidelines as well as the standard guidelines and principles of the Declarations of the Helsinki.

**Consent for publication.**   All study participants gave their consent to publish this work's findings.

## Statistical analysis

The experiment was created within Ngene (http://www.choice-metrics.com) which has been developed to created D-efficient designs which allows the researcher to force the design to maintain orthogonality while optimizing efficiency. Aggregate-level multinomial logit (MNL) analysis was executed to provide initial-level analysis of the choice data as was a basic count-analysis.

We analyzed choice data using correlated mixed multinomial logit models [43] to adjust for within-subject correlation [44, 45] and account for unobserved preference heterogeneity [43]. The model was estimated using the mlogit command with 500 random Halton draws in R version 4.1.2. No interaction terms were included and we did not perform sub-group analysis. In this analysis we applied dummy coding. For our main effects final model, we selected a mixed logit regression model to account for preference heterogeneity with all attributes included as random parameters.

In the mixed logit model, all the attributes were included as effects-coded categorical variables that we assumed to be normally distributed. This assumption was based on convenience because appropriate assumptions for these distributions remain ambiguous [27, 43, 46]. The level of each attribute that we expected to be most neutral was used as the omitted or reference attribute parameter. The negative preference (represented by negative coefficient) represents disutility or disliking of that option and positive preference represents liked or beneficial choice/utility (positive coefficient) on mixed logit model [S2 File].

## Results

### Response rate

A total of 192 participants who were registered in ANC clinics at Nairobi Metropolitan Service's Kariobangi and Kangemi health centers were contacted and screened for eligibility. Of these, 21 participants did not meet the eligibility criteria, where eight participants gave birth before the study commenced and 13 were aged more than 18 years. Out of 171 eligible participants, 18 refused to consent leaving a total sample size of 153 (Response rate of 89.5%) (see Fig 3).

**Participant sociodemographic characteristics.** Table 2 presents the socio-demographic and other characteristics of the respondents. A total of 153 pregnant adolescent girls participated in the survey. The mean age was 17.2 and ranged from 14–18 years. More than three-quarters of participants (79.7%) were single, while the rest were either married or living with a partner. In terms of education, the majority (72.5%) had secondary school level of education. Most respondents (64.1%) were students, and the rest were staying at home doing family chores. Participants who were classified as having probable depression (PHQ-9>10) were 43.1% (*95% C.I.* 35.3%–51.6%, mean (SD): 8.9 (9.0).

### Preference results

The results from the mixed logit model are presented in Table 3. Given the significant estimates for all but one attribute level information delivery and the alignment of coefficients with

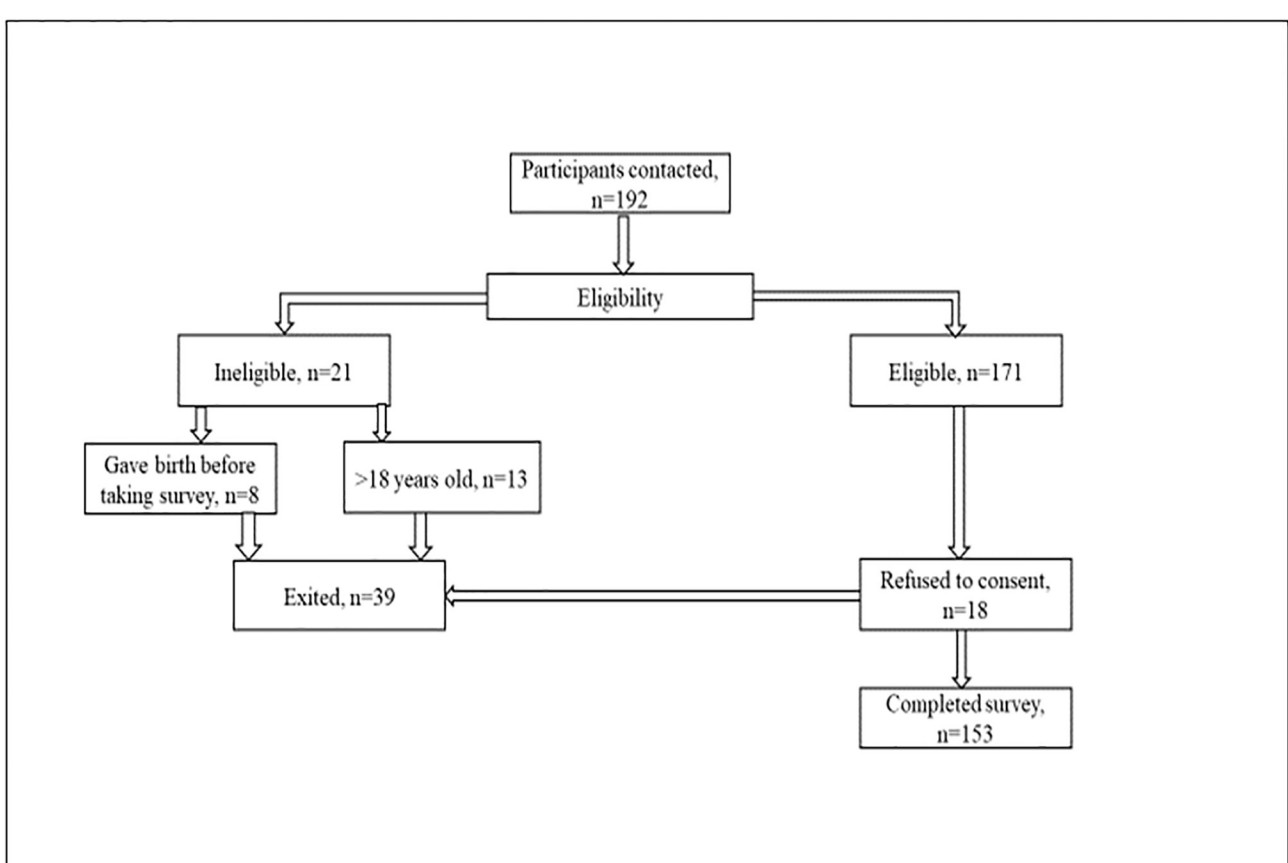

**Fig 3. Recruitment flow chart.**

**Table 2. Socio-demographic profile of the respondents.**

| Variable | Category | Frequency (N = 153) | Percentage (%) |
|---|---|---|---|
| Age | Mean; SD; Range | 17.2; 1.0; 14–18 | |
| Marital status | Single | 122 | 79.7 |
| | Married/ co-habiting with a partner | 31 | 20.3 |
| Highest level of education | Primary school | 42 | 27.5 |
| | Secondary school | 111 | 72.5 |
| Monthly Family Income | <4,999 KES/ under 500 USD | 82 | 53.6 |
| | 5,000–9,999 KES/500–100 USD | 34 | 22.2 |
| | >10,000 KES/1000 USD | 37 | 24.2 |
| Person Living With | Parents | 103 | 67.3 |
| | Spouse | 32 | 20.9 |
| | Others | 18 | 11.8 |
| Week of gestation of first ANC visit | < 12 weeks | 25 | 16.3 |
| | 12–28 weeks | 108 | 70.6 |
| | >28weeks | 20 | 13.1 |
| Unplanned Pregnancy | Yes | 96 | 62.7 |
| | No | 57 | 37.3 |
| Attitude of adolescent's partner towards the pregnancy | Positive | 92 | 60.1 |
| | Negative | 40 | 26.1 |
| | Ambivalent | 21 | 13.7 |
| Presence of social support | Yes | 120 | 78.4 |
| | No | 33 | 21.6 |
| Parity | 0 | 14 | 9.2 |
| | 1 | 113 | 73.9 |
| | 2+ | 26 | 17.0 |
| Age at Sex Debut | 14–16 Years | 79 | 52.0 |
| | 17–18 Years | 73 | 48.0 |
| | *Non-Response* | *1* | |

a priori expectations, we conclude the DCE was well understood by participants and the modelling method appropriate. The attributes are ranked from the most preferred to the least preferred in terms of the strength of their coefficients.

With regards to services delivery, the respondents had a lower preference of being offered services along with adult mothers at the ANC (63.7%, β = -37.92, *p<0.001*) as compared to separate adolescent friendly services. The respondents expressed higher preference for training in parenting skills (56.7%, β = 19.6, *p<0.001*) as compared to peer-support based skills. In terms of further training needs, there was a lower preference for return to school (77.3%, β = -19.3, *p<0.001*) as opposed to livelihood training.

In terms of added incentives to make improve access to IPT sessions, a lower preference for refreshments was expressed (51.6%, β = -4.69, *p<0.001*) as compared to provision of combined transport funds and refreshments. Interestingly, our respondents had a higher preference for receiving transport allowance (99.9%, β = 18.39, *p<0.001*) in comparison of both refreshments and travel allowance.

In terms of joint participation group sessions, the respondents showed a higher preference that the caregivers be provided with information sheets (64.4%, β = 6.05, *p<0.001*), as opposed to co-participation with caregivers or partners.

**Table 3. Results of mixed multinomial logit model with calculated proportions of positive and negative effects for treatment options.**

| Variable | Category | β | S. D | % Preference | S. E | Sig. |
|---|---|---|---|---|---|---|
| Information delivery | SRHR information | Ref. | | | | |
| | Individualized dietary information | -0.66 | 26.83 | -50.8 | 0.76 | 0.3864 |
| Joint Participation | Co-participate with Caregivers/partners | Ref. | | | | |
| | Provide information sheets for care-givers | 6.05 | 16.38 | 64.4 | 0.82 | <**0.001** |
| Treatment duration | 4 sessions for 1.5 hours | Ref. | | | | |
| | 8 sessions for 1.5 hours | 5.52 | 122.11 | 52.0 | 1.33 | <**0.001** |
| Intervention delivery agents | CHV | Ref. | | | | |
| | Facility Nurses | 6.04 | 87.78 | 52.8 | 1.27 | <**0.001** |
| Further Training Needs | Vocational | Ref. | | | | |
| | Formal (Back to school) | -19.29 | 25.77 | -77.3 | 1.13 | <**0.001** |
| Support type | Peer support | Ref. | | | | |
| | Parenting skills | 19.57 | 113.50 | 56.7 | 1.51 | <**0.001** |
| MCH Services | Adolescent friendly services | Ref. | | | | |
| | Combined with adult/older women | -37.92 | 107.82 | -63.7 | 1.64 | <**0.001** |
| Incentives | Food | -4.69 | 118.83 | -51.6 | 1.74 | **0.0071** |
| | Transport | 18.39 | 5.53 | 99.9 | 0.81 | <**0.001** |
| | Both | Ref. | | | | |
| Opt Out | | -118.10 | 32.58 | -99.9 | -118.10 | <**0.001** |

Note: Ref-Reference category; MCH-Maternal and child health care

When asked who is preferred for running intervention delivery sessions, the respondents had a higher preference for facility nurses (52.8%, β = 6.04, *p<0.001*) as compared to CHVs.

In terms of intervention delivery, our respondents showed a higher preference of 8 sessions (52.0%, β = 5.52, *p<0.001*) as compared to 4 sessions.

Participants did not show significant differences in preference for information delivery. Respondents were less likely to opt out of the choices with 99.9% of them opting to choose one of the two options provided. Meaning there was a very high intrinsic value to the service being offered (represented by the constant) so even a service with the worst level for each attribute is likely to be preferred to no service at all.

## Choice consistency

Choice consistency was measured by comparing respondent choices in the repeat choice set and the original choice set and counting the number of times the same alternative was chosen. The choice shares do not differ much between the test and the repeat test pattern (p = 0.415). Both alternative A and B are chosen slightly less in the repeat task 0.6% and 3.3% respectively. The opt-out share is slightly higher in the repeat choice (7.9%) than in the original choice (3.9%) but the share of respondents who consistently chose the opt-out is relatively low. 57.9% of the choices are consistent when comparing choice behavior between the original choice and the repeat choice.

## Ranking of attributes

Our respondents ranked Information delivery as the first, treatment duration option second, Support type third, Joint Participation fourth, MCH Services fifth, Incentives sixth, and Intervention Delivery Agents seventh and Further training needs as eighth (see Fig 4).

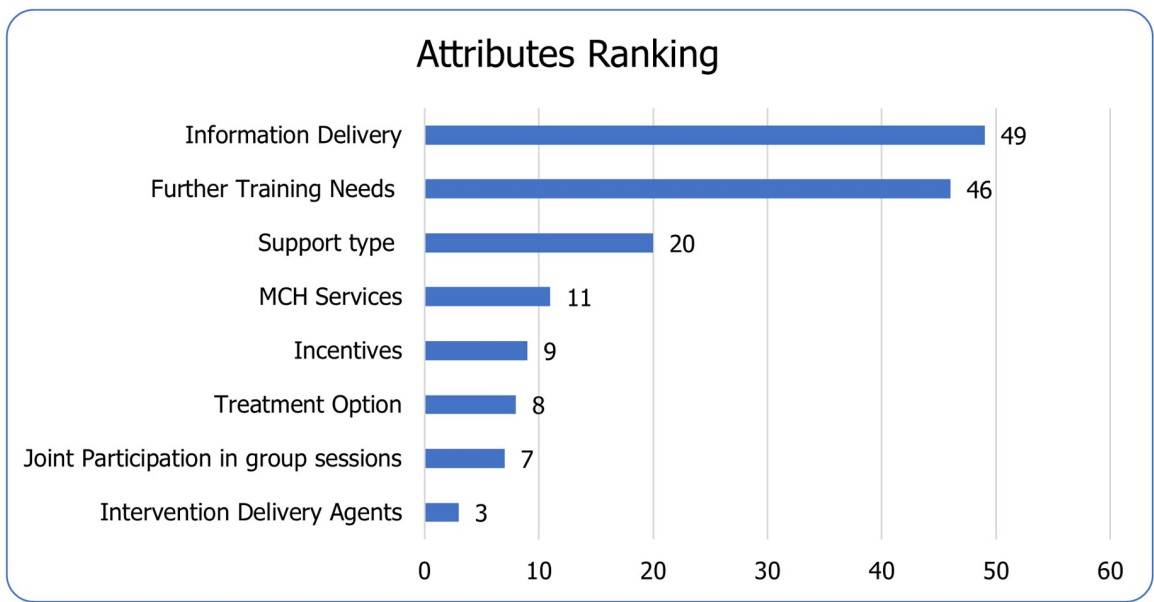

**Fig 4. Ranking of attributes.**

## Discussion

This study assessed depression treatment preferences among pregnant adolescent girls in informal urban setting. Consistent with prior expectations, information delivery and treatment options were the most important attributes. Our respondents showed a positive preference that caregivers be provided with information sheets, as opposed to co-participation in therapy sessions. With regards to treatment options, the respondents showed a positive preference for 8 sessions as compared to 4 sessions. With regards to intervention delivery agents, the respondents had a positive preference for facility nurses as compared to community health volunteers (CHVs). In terms of support, the respondents showed positive preference for parenting skills as compared to peer support. Respondents expressed negative preferences of ANC service combined with older mothers as compared to adolescent friendly services and of being offered refreshments alone. However, a positive preference was revealed for combined refreshments and travel allowance over travel allowance or refreshments alone.

These findings do suggest that young peripartum adolescents would prefer more tailored support that engages them directly but also provides guidance and engagement with their caregivers and partners. These engagements based on their preference are separate form direct engagement with them. There is a strong emphasis on youth friendly maternal and child health care services than being lumped with routine MCH clinics with adult women.

Interpersonal Psychotherapy (IPT) has been shown to have high efficacy when delivered by a readily available work force of local, non-specialist personnel [47–51] as demonstrated by recent studies in Kenya [52, 53]. Young pregnant women in Kenya experience a particular set of stressors which we believe will require adaptation of standard IPT in order to match patient needs and preferences. While IPT is well-equipped to address these risk factors, cultural and contextual adaptations are required to address specific role transitions and losses faced by young pregnant women. Building on coping skills that are consistent with local culture, including group support [53], we have proposed use of group Interpersonal psychotherapy (IPT-G), a well-established, protocol-driven format that can be delivered by non-specialists

and as mandated by the WHO [54], this adaptation would strengthen delivery of patient centered evidence-based interventions.

In general, DCEs have been shown as effective method for eliciting preferences for mental health services within diverse settings, illustrating a promising approach to increasing patient-centered mental health care.

These directly elicited preferences are consistent with our experience of depression associated challenges that pregnant adolescents experience [13–16]. We learnt that our respondents preferred longer group psychotherapy and that they did not rank educational needs above vocational training. Pregnancy and impending motherhood may have shaped these preferences which might evolve and it appears that they did see infant care and parenting as their key difficulties. Our respondents would prefer informational support for caregivers and partners above direct involvement of the caregivers and partners in group sessions. It appears that peer support, privacy and a safe space to share their experiences is considered important. Given that a large number of them experience interpersonal disputes with family members and are in conflicted relationship with their partners or partners, something other studies have noted too (45). It was also clear that they preferred to interact and be serviced in Antenatal clinics that are youth friendly/responsive by nurses but not with older adult women. Pregnant and parenting adolescents differentiate themselves from regular adolescents and adult women [55–57].

A preference of IPT delivery by ANC nurses over CHVs is also telling. While we will now use these revealed preferences as guidance in our efforts to further modify group IPT, we also know that as these young women give birth, their preferences may evolve and change. Keeping a conversation around other aspects of health such as robust SRHR choices- including family planning, use of PrEP, HIV testing, use of contraceptives etc. will also be critical and nurses and CHVs can both play a part there. Working out their livelihood options if return to school is difficult will need to be a priority and for those who will opt for continuation of school, offering brief IPT (4 sessions, even as a booster or remission treatment) might help in the long run. It appears that young pregnant or parenting girls would like to be taken seriously like adult women and would like to access services that are responsive to their needs and offer protection from multidimensional stigma associated with unintended early pregnancy and mental illness.

Depressive disorders are among the top three causes of years lived with disability globally, accounting for 40% of all mental illness, and affecting 350 million people comprising 4% of the population. Mental health services are scarce in low-and-middle-income countries like Kenya. Even when services exist, these do not map on to patient and provider preferences. Innovations are needed to provide accessible, affordable, and acceptable prevention, care, and treatment services to the diverse populations faced with poor mental health. Information and messages about mental health, preventative services, care and treatment characteristics, provider approaches, and care provision modalities must continue to evolve based on stakeholder preferences to ensure relevance and desirability [58].

Historically, patient involvement especially of vulnerable adolescents in shaping health practice has been minimal, especially in low-resource settings (10–13). There is good evidence that services that engage patients from the beginning—around conceptualizing the service itself, can be highly successful and effective. Similarly, adolescent-centered care has been associated with improved symptom burden, satisfaction, and enablement [59].

The current focus on patient-centered care within healthcare systems aims to ensure high-quality interactions between patients and the health system through achievement of the eight principles: *respect for patients' preferences, coordination and integration of care, information and education, physical comfort, emotional support, involvement of family and friends,*

*continuity and transition*, *and access to care* [60, 61]. It appears that many of these principles were articulated by our respondents in the DCE experiment on depression care.

## Limitations

This study had some limitations. It was conducted in two healthcare facilities in an informal urban setting that are part of Nairobi metropolitan services health facility and results may not be entirely generalizable to other settings and practice models. Therefore, the applications of our findings remain limited to urban informal settlements. These settlements tend to be socio-economically and ethnically diverse in their own right so the current study can inform more contextual study designs. We studied a convenience sample of respondents who may have been more frequent visitors to the facilities and their views may not represent all patients. Our results and conclusions are based on the attributes and levels included in the DCE we designed. While we followed a robust process to determine which attributes are important and relevant in our context using focus groups of key informants with expert knowledge of the clinical setting as well as previous literature in similar settings, we cannot be sure we captured all important attributes. While we did not look explicitly at preference heterogeneity by sub-group, we recommend that to future studies/ researchers to allow them identify the levels where there is preference heterogeneity. Despite the enormous significance of these findings, these are outputs of a DCE experiment from two Nairobi primary health care sites. This could be tested further in other sites and settings for external validity.

## Conclusion

Our participants revealed a complex set of preferences—prioritizing longer psychotherapy duration, parenting support, disseminating relevant depression care information to caregivers and partners as opposed to inviting them into groups, vocational training over return to school and combined refreshments with travel allowance as added incentives for psychotherapy. Negative preferences were revealed for combined ANC services with adult women, and provision of refreshments alone. These directly elicited preferences provide a unique opportunity to develop 'patient-centered' mental health services in a primary care context.

## Supporting information

**S1 File. Process checklist, indicators and key outcomes in developing DCE.**
(DOCX)

**S2 File. Supplementary mixed logit models.**
(DOCX)

**S3 File.**
(CSV)

## Acknowledgments

The authors would like to thank all the participants, Nairobi County health directorate, Director of Mental health, Ministry of Health, Kariobangi, and Kangemi health facility staff.

## Author Contributions

**Conceptualization:** Manasi Kumar.

**Data curation:** Manasi Kumar, Vincent Nyongesa.

**Formal analysis:** Manasi Kumar, Albert Tele, Matthew Quaife.

**Funding acquisition:** Manasi Kumar.

**Investigation:** Manasi Kumar, Joseph Kathono, Obadia Yator, Jurgen Unutzer.

**Methodology:** Albert Tele, Joseph Kathono, Jurgen Unutzer.

**Project administration:** Albert Tele, Joseph Kathono, Vincent Nyongesa, Obadia Yator, Shillah Mwaniga.

**Resources:** Manasi Kumar.

**Software:** Albert Tele, Matthew Quaife.

**Supervision:** Vincent Nyongesa, Obadia Yator, Shillah Mwaniga.

**Validation:** Manasi Kumar, Shillah Mwaniga, Matthew Quaife.

**Visualization:** Manasi Kumar, Albert Tele, Pim Cuijpers.

**Writing – original draft:** Manasi Kumar.

**Writing – review & editing:** Manasi Kumar, Albert Tele, Vincent Nyongesa, Obadia Yator, Keng Yen Huang, Mary McKay, Joanna Lai, Marcy Levy, Pim Cuijpers, Matthew Quaife, Jurgen Unutzer.

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
