## [Decision Letter · Decision Letter 0]

19 Oct 2022

PONE-D-22-21937A discrete choice experiment to understand depression intervention treatment preferences of Kenyan pregnant adolescentsPLOS ONE

Dear Dr. Kumar,

Thank you for submitting your manuscript to PLOS ONE. After careful consideration, we feel that it has merit but does not fully meet PLOS ONE’s publication criteria as it currently stands. Therefore, we invite you to submit a revised version of the manuscript that addresses the points raised during the review process. This is a very important article and topic, thank you for writing and doing this research. Please review the comments and use them to revise and strengthen your paper. The feedback provides complementary information with one reviewer focusing more heavily on the statistical methods used and one on more broader comments to increase the overall clarity of the article.  The article needs revision, the only options were minor or major, so erred up as there is a considerable amount of feedback to respond to, and I look forward to reviewing again.  

Please submit your revised manuscript by Dec 03 2022 11:59PM. If you will need more time than this to complete your revisions, please reply to this message or contact the journal office at plosone@plos.org. Please include the following items when submitting your revised manuscript:

We look forward to receiving your revised manuscript.

Kind regards,

Melissa Sharer

Academic Editor

PLOS ONE

Journal Requirements:

7. Please ensure that you refer to Figure 3 and 4 in your text as, if accepted, production will need this reference to link the reader to the figure.

Reviewers' comments:

Reviewer's Responses to Questions

**Comments to the Author**

1. Is the manuscript technically sound, and do the data support the conclusions?

Reviewer #1: Yes

Reviewer #2: Yes

2. Has the statistical analysis been performed appropriately and rigorously? 

Reviewer #1: Yes

Reviewer #2: Yes

3. Have the authors made all data underlying the findings in their manuscript fully available?

Reviewer #1: Yes

Reviewer #2: Yes

4. Is the manuscript presented in an intelligible fashion and written in standard English?

Reviewer #1: Yes

Reviewer #2: Yes

5. Review Comments to the Author

Reviewer #1: This was an interesting discrete choice experiment which sought to estimate the preferences of pregnant adolescents for depression services in Kenya. The paper was generally clear and the description of attribute and level selection was particularly thorough. I also found the discussion section interesting to read. For publication there are a number of areas where I believe the analysis has been conducted well but could be more clearly reported in the manuscript.

Major comments:

1. The authors should take care in interpreting the coefficients in the model. As all the attributes are dummy coded the coefficients represent preferences for a service with that attribute compared to whichever attribute was dropped in the analysis. Where a level has a negative coefficient this does not necessarily mean that the level was disliked, only that the other level in that attribute was preferred to it. There was a very high intrinsic value to the service (represented by the constant) so even a service with the worst level for each attribute is likely to be preferred to no service at all.

2. Given the use of dummy variables, it would be helpful to note the interpretation of the constant in the text. This will be the value of a service containing each of the dropped levels compared to no service.

3. The authors state in the manuscript that the variables are effects coded but the supplementary appendices and results table suggest they are dummy coded. On Page 8 Line 197 the authors state that they use effects coding to deal with non-linearities which appears to be incorrect and irrelevant given that all of the levels are categorical not contiuous.

4. The abstract states that the authors used an orthogonal design but this only seems to have been for the pilot which was then used to generate priors for a Bayesian d-efficient design. This should be corrected in the abstract. The use of a Bayesian d-efficient design will also mean that the coefficients for the levels are not estimated independently of each other and are unlikely to be balanced so this should be changed in the manuscript.

5. Could the authors clarify whether an uncorrelated or correlated mixed logit model was estimated? Only the correlated mixed logit accounts for correlations such as scale heterogeneity. Could the authors also clarify whether they tested any other models such as the conditional logistic regression model?

6. While the authors do describe the meaning of the levels in the supplementary materials it would be useful to potentially add a brief description of these to table 1 in the main manuscript. There is also currently no reference to the level descriptions in the supplementary materials in the text.

7. The authors included a test for consistency of preferences but didn't then analyse the results of this. Could they add this to the manuscript?

8. While the authors don't explicitly look at preference heterogeneity by sub-group, the mixed logit should allow the researchers to identify the levels where there is preference heterogeneity. IT would be useful to comment on these for future research

Minor comments:

P4 L103 : DCEs don't ask for utilities, they allow them to be estimated from participants' choices

P4 L109 : Participants choose their most preferred service rather than favourite. It might be that in a given question they have to choose the lease negative option.

P7 L168 : The year the survey was conducted is incomplete

P12 L276 : I think saying the enormous significance of these results is overselling the findings slightly

Reviewer #2: A discrete choice experiment to understand depression intervention treatment preferences of Kenyan pregnant adolescents

This is a novel research topic on a very important issue. Mental health is increasingly highlighted as a global concern, as are adolescents who are pregnant or parents in LMICs. There is very little research which engages young people on preferences for treatment and support – and this is excellent to see.

I would recommend this for publication, with some suggested edits:

1. Some of the wishes of participants go beyond depression intervention treatment – for example the desire for specialised ANC services distinct from adult services. Perhaps this could be reflected a bit more in the title and abstract? It is certainly coming into the discussion.

2. I would add into the introduction that there has been a lack of research that includes and asks pregnant adolescents what they would want from such services – at least quantitatively – and cite any other studies that attempt this. A bit of review of existing evidence on adolescent preferences for mental health services in this introduction stage would be helpful.

3. Nice clear description of methods. Excellent way of compiling the list of choices, including qualitative and pilot stages. Very robust.

4. A bit more information about the purposive sampling would be useful here. In what ways was it purposive? Were all registered adolescents approached or was there purposive sub-sampling within this group?

5. Stable mental health was a criteria – please unpack this a bit more – what was meant by this?

6. Results very clearly and well written up.

7. Discussion – starts with ‘consistent with prior expectations’ – is this based on literature review of prior studies of preferences? Again this would be helpful to include.

8. The limitations are mentioned in two separate paragraphs apart from each other – it would help the flow to move them together

9. Use of this study to modify and improve group IPT is mentioned – it would be good to include this context in the introduction

10. There are some parts of the discussion which may be valuable to think about including in the introduction rather – for example setting context on patient involvement.

Overall an excellent and valuable study. It would benefit from a bit of restructuring between discussion and introduction, and some literature review of existing work – but these should be easily achieved. Well done to the author and team.

6. PLOS authors have the option to publish the peer review history of their article (what does this mean?). If published, this will include your full peer review and any attached files.

Reviewer #1: **Yes: **Stuart Wright

Reviewer #2: **Yes: **Professor Lucie Cluver

---

## [Decision Letter · Decision Letter 1]

27 Jan 2023

Understanding depression treatment and perinatal service preferences of Kenyan pregnant adolescents: a discrete choice experiment

PONE-D-22-21937R1

Dear Dr. Kumar-

We’re pleased to inform you that your manuscript has been judged scientifically suitable for publication and will be formally accepted for publication once it meets all outstanding technical requirements.

Kind regards,

Melissa Sharer

Academic Editor

PLOS ONE

Reviewers' comments:

Reviewer's Responses to Questions

**Comments to the Author**

1. If the authors have adequately addressed your comments raised in a previous round of review and you feel that this manuscript is now acceptable for publication, you may indicate that here to bypass the “Comments to the Author” section, enter your conflict of interest statement in the “Confidential to Editor” section, and submit your "Accept" recommendation.

Reviewer #1: All comments have been addressed

2. Is the manuscript technically sound, and do the data support the conclusions?

Reviewer #1: Yes

3. Has the statistical analysis been performed appropriately and rigorously? 

Reviewer #1: Yes

4. Have the authors made all data underlying the findings in their manuscript fully available?

Reviewer #1: Yes

5. Is the manuscript presented in an intelligible fashion and written in standard English?

Reviewer #1: Yes

6. Review Comments to the Author

Reviewer #1: (No Response)

7. PLOS authors have the option to publish the peer review history of their article (what does this mean?). If published, this will include your full peer review and any attached files.

Reviewer #1: **Yes: **Stuart Wright

---

## [Editor Report · Acceptance letter]

27 Feb 2023

PONE-D-22-21937R1 

Understanding  depression  treatment and perinatal service preferences of Kenyan pregnant adolescents: a discrete choice experiment 

Dear Dr. Kumar:

I'm pleased to inform you that your manuscript has been deemed suitable for publication in PLOS ONE. Congratulations! Your manuscript is now with our production department. 

Kind regards, 

on behalf of

Dr. Melissa Sharer 

Academic Editor

PLOS ONE